# Effect of SiO_2_–Al_2_O_3_ Glass Composite Coating on the Oxidation Behavior of Ti60 Alloy

**DOI:** 10.3390/ma13225085

**Published:** 2020-11-11

**Authors:** Wenbo Li, Ken Chen, Lanlan Liu, Yingfei Yang, Shenglong Zhu

**Affiliations:** 1State Grid Hunan Electric Power Company Limited Research Institute, 388 ShaoShan N Rd, Changsha 410000, China; liwb@alum.imr.ac.cn; 2Laboratory for Corrosion and Protection, Institute of Metal Research, Chinese Academy of Sciences, 52 Wencui Rd, Shenhe District, Shenyang 110016, China; slzhu@imr.ac.cn; 3School of Materials Science and Engineering, Dongguan University of Technology, 1 Daxue Rd, Dongguan 523808, China; chenk@dgut.edu.cn; 4Live Inspection and Intelligent Operation Technology State Grid Corporation Laboratory, State Grid Hunan Transmission Maintenance, 8 Lixiang M Rd, Changsha 410000, China; 0603050307@163.com; 5Institute of Advanced Wear & Corrosion Resistant and Functional Material, Jinan University, 601 Huangpu W Ave, Tianhe District, Guangzhou 510632, China

**Keywords:** titanium alloy, oxidation, glass-ceramic, coating, interfacial reaction

## Abstract

A SiO_2_–Al_2_O_3_ glass composite coating was prepared on Ti60 alloy via air spraying slurry and then a suitable baking process. It was composed of potassium silicate glass, alumina and quartz powders. The high temperature oxidation performance of the alloy with and without coating was evaluated in static air at both 800 °C and 900 °C. The results show that catastrophic oxidation occurs for Ti60 bare alloy. It had a mass gain of about 2 mg/cm^2^ after oxidation at 800 °C and 17 mg/cm^2^ at 900 °C for 100 h. On the contrary, the oxidation resistance of alloy coated with composite coating was much improved with the mass gain about 0.36 mg/cm^2^ and 0.95 mg/cm^2^ at 800 °C and at 900 °C, respectively. The microstructure evolution of the composite coating and the alloy was analyzed by scanning electron microscope and electron probe microanalyzer. The effect of the composite coating on the oxidation performance of the alloy is discussed especially in terms of oxygen diffusion and interfacial reaction.

## 1. Introduction

Titanium alloys are attractive structural materials for their high specific strength, excellent corrosion resistance and stable moderate temperature properties [1]. They have been widely used in areas of aerospace, automotive, military, sports equipment and chemical engineering [1,2,3,4,5,6]. However, their extensive practical applications are limited by the poor high-temperature oxidation resistance. This is because a TiO_2_/TiO_2_ + Al_2_O_3_ layer, which is porous and less protective, will form during high temperature exposure. To improve the high-temperature oxidation resistance of the titanium alloys, a protective coating is often applied in industry. Among these coatings, the glass-ceramic coating attracts much attention [7,8,9,10,11,12].

In general, traditional glass-ceramic coatings are usually prepared by a firing-crystallization method [13,14,15,16,17]. This coating preparation process can be divided into the following steps: (1) mixing the glass powder sufficiently; (2) melting and quenching the mixed powder to obtain the original glasses with given composition; (3) milling the glasses into given particle size; (4) spraying the glasses powder to the sample and then baking at high temperature; (5) heat treatment at crystallization temperature.

In order to get good thermal and mechanical properties, the types of crystals in the glass should be well controlled, that is, to avoid the precipitation of the coarse acicular-shaped crystals [8]. This requires designing the glass composition and heat treatment elaborately. Unfortunately, both of them are hard to control accurately. As a result, a large number of nucleating agents (usually ZrO_2_, CaF_2_, etc.) are usually added into the glass for the sake of achieving suitable crystals [18,19,20]. The nucleating agents will significantly change the properties of the glass, such as the thermal expansion coefficient, softening temperature, and so on. However, it will further complicate the coating preparation. In order to achieve the optimized mechanical property of the glass-ceramics, eight ingredients were added in mixture of the glasses in Fu’s work [20,21]. Moreover, with the increase of complexity in composition, the crystallization of the glasses is changed. Hence, a series of heat treatments is required [22]. The difficulty with heat treatments also lies in accurate control of the temperature for the complex-shaped workpiece. For example, the heating rate in the thicker region is lower than that in the thinner region, so that the crystallization of the coating is inhomogeneous.

A novel method of preparing glass-ceramic coating has been proposed in our previous work [23,24]. In this method, a slurry is made by mixing the desired ceramic particles with binary alkali silicates solution directly. Furthermore, the small colloids particles in the solution of binary alkali silicates aggregate and form the glass matrix during the baking process. In this way, the microstructure of the glass-ceramic coatings is uniform. Moreover, it can be prepared very conveniently and cheaply.

In considering the excellent oxidation resistance of α–Al_2_O_3_ or SiO_2_ scales, they are added as the crystalline phase in the glass-ceramic coating in the current work [25,26]. The oxidation kinetics and microstructure evolution of the composite coating are analyzed at 800 °C and 900 °C on Ti60. Besides, attention is paid to observation of the interfacial reaction and interface evolution at the coating/substrate interface. The effect of the composite coating on the oxygen diffusion during high temperature exposure is discussed for the first time.

## 2. Experimental Procedures

### 2.1. Coating Preparation

Ti60 alloy with the nominal composition shown in Table 1 is used as the substrate. The substrate is constituted by an ordered α phase and a disordered α_2_ phase. Specimens with the dimension of 15 × 10 × 2.5 mm^3^ were cut by a spark-discharging machine. The surface treatments of the raw specimens included grinding down to a final 600# SiC sandpaper and then blasting with alumina (300 mesh in size). The degreasing was conducted in ethanol ultrasonically for 10 min. The concentration of the aqueous solution of potassium silicate (ASPS) is 40 wt.% with the ratio of SiO_2_ to K_2_O about 3:1 (Dayang Chemical Plant, Beijing, China). The sizes of α–Al_2_O_3_ (Laisheng Plant, Shenyang, China) and quartz (Sinopharm Chemical Reagent Co., Ltd., Shanghai China) powders were ranged 1–10 μm. The detailed preparation process of SiO_2_–Al_2_O_3_ glass composite coating can be described as follows: Firstly, prepared the slurry by mixing α–Al_2_O_3_ and quartz particles into ASPS. The composition of the slurry is 20 wt.% quartz, 7 wt.% α–Al_2_O_3_ and 72 wt.% ASPS. In order to make the particles homogeneous, the mixture was milled via ball milling for 15 min. The speed of the ball milling is 1500 r/min. Secondly, the slurry was sprayed on the substrate with a spraying gun. The distance between the sample and muzzle was about 20 cm. The spraying pressure was set at 0.4 MPa. The spraying was sustained for 10 s each time and repeated for 15 times together. Finally, the coated sample was solidified and gradually baked according to the schedule shown in Figure 1a. This baking schedule insures the water in the coating departs slowly. It avoids the formation of cracks in the coating during baking, which can be proved by the surface morphology in Figure 1b.

### 2.2. Oxidation Tests

The oxidation test of the composite coating was conducted in a batch-type muffle furnace (Kere Furnace Co., Ltd., Luoyang, China) both at 800 °C and 900 °C. For comparison, the bare Ti60 alloy was also oxidized simultaneously. After being oxidized for a certain time, the samples were removed out from the furnace quickly and cooled in the air. An electronic balance (Sartorius BP211D, Wood Dale, IL, USA) with the sensitivity of 10^−5^ g was used to record the average mass gain at the oxidation intervals. Three parallel samples were tested in order to get the accurate mass change. The weight of the samples was measured with together the alumina crucible. This ensures the spalled oxide was part of the mass gain. Before the oxidation test, the crucibles were preheated at 1200 °C for sufficient time until no further weight change was observed.

### 2.3. Analytical Characterization

The surface and cross-sectional morphologies of the specimens after oxidation were characterized by field-emission scanning electron microscope (SEM, Inspect F 50, FEI Co., Hillsboro, OR, USA). An energy dispersive spectrometer (EDS, OXFORD X-Max, Oxford Instruments, Oxford, UK), which is equipped with SEM, was used to determine the composition of the coating and interface. The phase constitution was detected by X-ray diffraction (Panalytical X’ Pert PRO, Cu Ka radiation at 40 KV, PA Analytical, Almelo, The Netherlands). The elemental distribution of the composite coating, coating/substrate interface and substrate after oxidation was studied by electron probe microanalysis (EPMA-1610, Shimadzu, Kyoto, Japan).

## 3. Results

The coating after baking is compact without any visible cracks and voids according to Figure 1b. The mass gain of Ti60 alloys with and without the SiO_2_–Al_2_O_3_ glass composite coating during oxidation exposure at 800 °C and 900 °C is shown in Figure 2. It can be noted from Figure 2a that the bare Ti60 shows rapid weight gain during the whole oxidation test at 800 °C. Its total mass gain after oxidation exposure for 100 h is about 2 mg/cm^2^. On the contrary, the oxidation rate for alloy with composite coating is much lower. The weight gain of the coated sample is only 0.36 mg/cm^2^. A novel phosphate-ceramic coating was developed for high-temperature oxidation protection of Ti65 titanium alloy by Han [27]. The isothermal oxidation test at 650 °C showed that the phosphate-ceramic coating had a mass gain of 0.35 mg/cm^2^ after oxidation for 20 h. It is well accepted that the oxidation rate of the coating increases with the rise of the exposure temperature. However, the fact is that the composite coating in the current work showed a similar oxidation rate with phosphate-ceramic coating even exposure at higher temperature. Therefore, the composite coating here possesses superior oxidation resistance than phosphate-ceramic coating. When the oxidation temperature is elevated to 900 °C, the mass gain for both samples increases significantly. As it is shown in Figure 2b, the mass gain for bare Ti60 grows in a straight line. The total mass gain is as large as 17 mg/cm^2^ after oxidation for 100 h, indicating the occurring of catastrophic oxidation. The mass gain of the coated alloy after oxidation at 900 °C was more than doubled when compared with that at 800 °C. However, it was still much lower than the bare Ti60 alloy. Moreover, an approximate parabolic growth curve remains for the coated alloy. This hints that satisfactory protection has been achieved on the composite coating. The XRD patterns of the composite coating before and after oxidation test is shown in Figure 3. According to the results, quartz, cristobalite, α–Al_2_O_3_, Ti_5_Si_3_ and Ti_3_Al are detected in the composite coating before oxidation. Combining the coating preparation process, the quartz and α–Al_2_O_3_ are the inclusions. Instead, neither Ti_5_Si_3_ nor Ti_3_Al has been added in the original ingredient, but their patterns are intense and obvious. This indicates that reactions have occurred at the coating/alloy interface during baking. Intermetallic phases of Ti_5_Si_3_ and Ti_3_Al are the products.

After oxidation, the intensity of Ti_5_Si_3_ and Ti_3_Al peaks becomes stronger evidently, which displays that the coating/alloy interfacial reaction continues during oxidation. It is intriguing to notice that cristobalite, a polymorph of the silica, is only detected after oxidation at 900 °C. According to Holmquist [28], cristobalite is easier to form at high temperature. However, cristobalite is not found after oxidation at 800 °C. This indicates that the interfacial reaction only occurs when the temperature is high enough, which agrees with Holmquist’s results well [28].

Figure 4 shows the surface and cross-sectional morphology of bare Ti60 substrate after oxidation at 800 °C for 100 h. From Figure 4a, it can be noted that the oxide formed on bare Ti60 is tetragonal with the size less than 2 μm, which is a typical morphology of TiO_2_. Besides, according to the Ti–O phase graph [29], TiO_2_ is the most stable phase under oxidation at 800 °C. The oxide scale is about 7 μm in thickness and porous in morphology. Furthermore, an obvious multilayer structure is found. The interfaces are tagged out by the red arrows in Figure 4b. This multilayer structure hints that the adherence of the oxide scale is poor. Serious oxide spallation will occur when the oxidation test continued, or the oxidation temperature is further elevated.

Typically, a protective oxide scale should be compact, continuous, adherent, and slow growing [25]. Therefore, the oxide formed on bare Ti60 hardly gives any protection for the alloy. Catastrophic spallation of oxide scale occurs after oxidation at 900 °C for 100 h, which can even be identified by the naked eye (Figure 5a). As a result, it is difficult to prepare cross-sectional samples for SEM observation. Instead, the surface of the coated sample is relatively intact (Figure 5b). Serious oxidation is only found around the hanging hole. This is because the coating around the hanging hole is inhomogeneous.

The surface and cross-sectional morphology of the coated specimens after oxidation at 800 °C and 900 °C are shown in Figure 6. The protrusions are observed at the surface of the coating. They are speculated to be Al_2_O_3_ and quartz inclusions by combining the XRD results. After oxidation at 800 °C for 100 h, the protrusions are fine and distribute homogeneously. Instead, when the oxidation temperature is elevated to 900 °C, the protrusions aggregates and grows up with a much larger volume. It is worth noting that neither micropores nor microcracks are observed at the surface under SEM, which means the coating is compact and possesses satisfactory barrier to the corrosive environment. 

The cross-sectional morphologies in Figure 6b,d display the details of the composite coating after oxidation. It is noticeable that the boundaries of quartz/glass are blurry while they are apparent for the alumina/glass boundaries. The interface of coating/alloy (pointed by red arrows) after oxidation at 800 °C and 900 °C is different. For oxidation at higher temperature, the interface is rougher. Furthermore, the coating becomes thinner accordingly. This phenomenon suggests that the interfacial reaction at 900 °C is more vigorous.

Elemental distribution of the coating/substrate interface and its nearby zones was analyzed by EDS linear scan (Figure 7). The direction of the arrow is consistent with direction of the line scan. It gives more information about the interface evolution during the oxidation exposure. As elements composition changed significantly at the interface, it can characterize the area of interface zone (as pointed out by the dashed line in Figure 7c,d). In Figure 7a,b, it is clear a multilayered interface has formed between the coating and alloy. The interfacial reaction zone is composed of two different sublayers after oxidation at 800 °C. The first layer is rich in Ti and Si. It corresponds to Ti_5_Si_3_ detected by XRD. Dispersed oxides and voids are observed in this layer (Figure 7a). The inside layer has a high concentration of Ti and Al. It is Ti_3_Al layer combining the XRD patterns.

For the specimens after oxidation at 900 °C for 100 h, the interfacial reaction zone is about 15 μm in thickness. It is much thicker than that after oxidation at 800 °C (about 6 μm). Similarly, Ti_5_Si_3_ and Ti_3_Al layer forms within the reaction zone. However, a sharp Al peak was also observed between the composite coating and Ti_5_Si_3_ layer (as pointed by the arrow in Figure 7d). It means a thin Al-rich layer has formed at higher oxidation temperature. It is possible that it is alumina but more proof is needed. 

Elemental mapping of coating/alloy interface after oxidation at 900 °C for 100 h by EPMA is shown in Figure 8. According to the distribution of Ti, Al and Si, Ti_3_Al and Ti_5_Si_3_ layers are confirmed. The Al distribution between the composite coating and Ti_5_Si_3_ layer is continuous. Oxygen distributed uniformly in the coating. An O-rich zone exists between the composite coating and Ti_5_Si_3_ layer, which is right consistent with the Al-rich zone. This result verifies that Al in the Al-rich layer is in the form of alumina. A trace amount of Al is also detected in the glass matrix, suggesting that the Al_2_O_3_ particles may partially dissolve into glass matrix [24]. No oxide of Ti is found in the composite coating, which means that elements from the substrate are constrained beneath the coating/alloy interface. 

## 4. Discussion

### 4.1. Effect of the Composite Coating on the Oxidation Resistance of Bare Ti60 Substrate

Based on the above results, the Ti60 alloys with SiO_2_–Al_2_O_3_ glass composite coating shows much better oxidation resistance when compared to bare Ti60 alloy. It hints that the composite coating possesses satisfactory oxidation performance at high temperature. It can significantly improve the oxidation resistance of the Ti60 coating. The following parts will focus on the oxidation mechanism of the composite coating.

The superior oxidation resistance of the composite coating is mainly attributed to two aspects. Firstly, the diffusion rate of oxygen in the composite coating is low. This means that the composite coating acts as a diffusion barrier for oxygen during the oxidation exposure. It is widely accepted that the diffusion coefficient of oxygen in silicate glass, alumina and silica is relatively low [30]. When these ingredients mix together to form the composite coating, a synergistic effect on preventing oxygen diffusion results [31]. 

The oxidation process contains the following steps [32]: (1) oxygen in the environment diffusing to the O_2_/MO (means oxide scale) interface and being absorbed there; (2) oxygen molecule being ionized into O^2−^ at the interface and the metal being ionized into M^2+^ also; (3) O^2−^ reacting with M^2+^ to from MO; and (4) O^2−^ diffusing through oxide scale to reach the MO/M interface and/or M^2+^ diffusing through oxide scale to reach O_2_/MO interface. Step (4) is necessary to sustain the growth of the oxide scale.

Typically, the oxidation rate is determined by the slowest step above. For example, for the vacuum system, there is little O can be absorbed, the oxidation rate of the metal is determined by step (1). For oxidation at high temperature, step (2) and (3) is fast, the oxidation rate is controlled by the diffusing rate of the ions through the oxide scale.

For the Ti60 alloys, a thin TiO_2_ can form fast in the first a few seconds by steps (1)–(3). However, there is a high oxygen solubility in TiO_2_, which entitles a fast O^2−^ diffusing rate through the oxide scale [25]. Therefore, the oxidation rate of Ti60 is fast during the whole test. In comparison, the diffusion rate of oxygen in the composite coating is low. When the alloy is coated with the composite coating, step (1) is much slower. This leads to a really low oxygen pressure at the coating/alloy interface. As a result, only alumina can be formed by selective oxidation [32]. Moreover, the oxygen solubility in Al_2_O_3_ is much lower than that in TiO_2_ [25], which slows down step (4). Therefore, the oxidation rate for the coated alloy is much lower.

The satisfactory oxidation resistance of the coating is also benefits from the excellent adherence of the coating to the alloy. This is because the coating/alloy interfacial reaction results in a good chemical bonding between coating and alloy. Moreover, the coefficient of the thermal expansion (CTE) of the titanium alloys is about 10–12 × 10^−6^ (K^−1^) [33]. The CTE of composite coating is estimated to be about 13.4 × 10^−6^ (K^−1^) by the Kerner’s model in our previous study [24]. The small difference in CTE of the coating and alloy ensures their interface with good resistance to spallation during repeated heating and cooling process. Therefore, the composite coating provides effective protection for Ti60 during the whole high temperature exposure. 

### 4.2. Interfacial Reaction between the Composite Coating and Ti60 Substrate

The interfacial reaction occurs between the coating and alloy during the oxidation tests. It leads to the formation of a multilayered interface zone at the coating/alloy interface. The products of the interfacial reactions are Ti_5_Si_3_ and Ti_3_Al. When the oxidation temperature is elevated to 900 °C, Al_2_O_3_ also forms. The following parts will focus on discussing the mechanisms of forming the multilayered interface.

As TiO_2_ is more thermodynamically stable than SiO_2_, the Si atoms in SiO_2(glass)_ are replaced by Ti atoms during high temperature [34,35,36,37,38]. This reaction releases some free Si atoms. In addition, a big difference exists in Si content between the coating and alloy. Therefore, the free Si atoms diffuse easily towards the coating/alloy interface driven by the concentration gradient. Similarly, Ti atoms also diffuse towards the interface. Hence, the intermetallic of Ti–Si, such as Ti_5_Si_3_, formed at the coating/alloy interface [37]. However, though the reactions between titanium and glass are widely investigated, no agreement has been reached in the mechanisms. One of the most accepted reactions between them is shown as reaction (1):8Ti + 3SiO_2_ = 3TiO_2_ + Ti_5_Si_3_(1)
where Ti alloy reacts with SiO_2_ directly to form Ti_5_Si_3_. In this study, a thin Al_2_O_3_ layer, besides the Ti_5_Si_3_ and Ti_3_Al, has been observed at the interfacial reaction zone at 900 °C. Considering the possible reactants, the following reaction (2) may also occur:5Ti+4Al+3SiO_2_ = Ti_5_Si_3_+2Al_2_O_3_(2)

With the formation of Ti_5_Si_3_ layer, Al is enriched at the coating/ Ti_5_Si_3_ interface. Moreover, oxygen partial pressure at the interface is low due to the slow oxygen diffusion rate in the composite. Hence, Al_2_O_3_ is favored to be selectively oxidized at such low oxygen pressure. The standard Gibbs free energy changes (ΔG^θ^) of forming Ti_5_Si_3_ according to Reactions (1) and (2) is shown in Figure 9 (standardized to forming 1 mol Ti_5_Si_3_). According to Figure 9, the ΔG^θ^ of both reactions is negative, indicating that the formation of Ti_5_Si_3_ is thermodynamically feasible. The value of ΔG^θ^ reflects the potential of the reactions and the stability of the products. It is worth noting that the ΔG^θ^ value for reaction (2) is more negative than that for reaction (1). This means that reaction (2) is thermodynamically more favorable.

On the other hand, the Al content in Ti_5_Si_3_/ alloy interface is also increasing gradually with the formation of Ti_5_Si_3_. The solubility of Al in Ti_5_Si_3_ phase is low. As a result, Ti_3_Al forms when the Al content exceeds its solubility. To better understanding the interface reactions, the schematic diagram showing this process is shown in Figure 10. The interfacial reaction can be divided into three steps: firstly, free Si atoms gather at the original coating/alloy interface due to replacement of Si by Ti in SiO_2_ (Figure 10a); then a thin Ti_5_Si_3_ layer forms. Simultaneously, Al is expelled from the Ti_5_Si_3_ lattice and enriches at both sides of Ti_5_Si_3_ layer (Figure 10b); finally, Al atoms at the coating/Ti_5_Si_3_ interface are selectively oxidized to form Al_2_O_3_, and Al atoms at the Ti_5_Si_3_/alloy interface react with Ti to form the Ti_3_Al layer (Figure 10c).

### 4.3. The Developing Trends of SiO_2_–Al_2_O_3_ Glass Composite Coating

Based on the above results and discussion, the SiO_2_–Al_2_O_3_ glass composite coating can provide satisfactory protection for the Ti60 alloy even when the oxidation temperature is as high as 900 °C. However, there is still a long way to go before its engineering application. The developing trends of the SiO_2_–Al_2_O_3_ glass composite coating is mainly focused on the following aspects:To optimize the composition and structure of the coating, e.g., by adjusting the ratio of SiO_2_ to K_2_O, optimizing the distribution, content and the size of the ceramic particles. These are ways to further lower the diffusion rate of oxygen in the coating and to achieve further improved oxidation resistance.To investigate the corrosion behavior and mechanisms of the composite coating under molten salts as well as environments with high Cl^−^ content.To adequately understand the reactions of each components at high temperature and seek more economical components, such as kaolin, which makes the composite coating oxidation-resistant and low-cost.

## 5. Conclusions

A novel SiO_2_–Al_2_O_3_ glass composite coating was successfully prepared on Ti60 alloy. The oxidation performance of the coating was evaluated at 800 °C and 900 °C. Based on the experimental results, the following conclusions could be drawn: The composite coating exhibits a dense structure after baking. It is adherent to the Ti60 substrate after oxidation due to their similar CTE and occurring of interfacial reaction at the coating/substrate interface.A much better oxidation resistance is achieved on Ti60 alloy with composite coating. The mass gain of the coated alloy is about a half and to a fifth of the bare alloy after oxidation for 100 h at 800 °C and 900 °C, respectively.Coating/alloy interfacial reactions occur during oxidation tests and a multilayered structure forms. This complex structure includes composite-coating/Al_2_O_3_-layer/Ti_5_Si_3_-layer/Ti3Al-layer/substrate.

## Figures and Tables

**Figure 1 materials-13-05085-f001:**
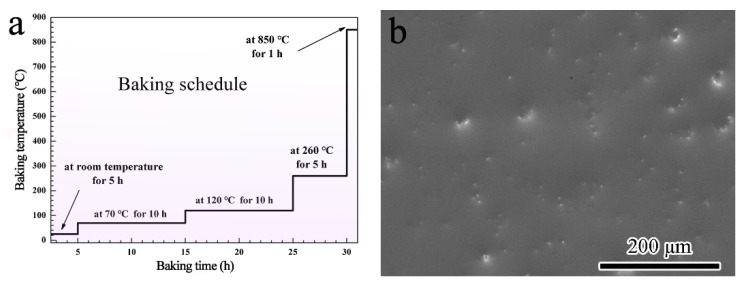
The baking schedule of the composite coating (**a**) and surface morphology of the coating after baking (**b**).

**Figure 2 materials-13-05085-f002:**
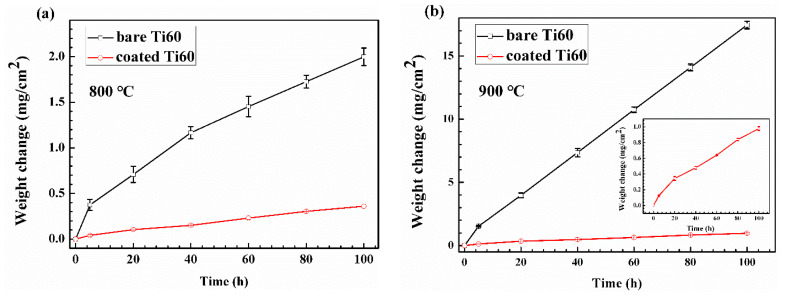
Mass change curve during oxidation at (**a**) 800 °C and (**b**) 900 °C.

**Figure 3 materials-13-05085-f003:**
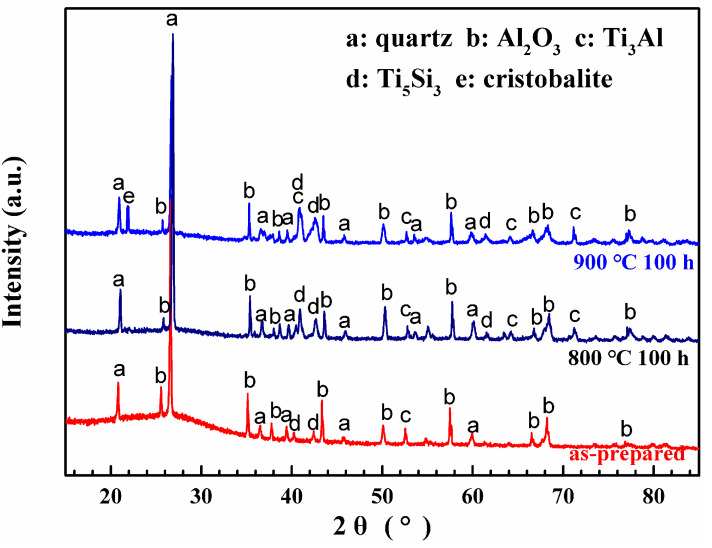
XRD patterns of the composite coating as prepared and after oxidation.

**Figure 4 materials-13-05085-f004:**
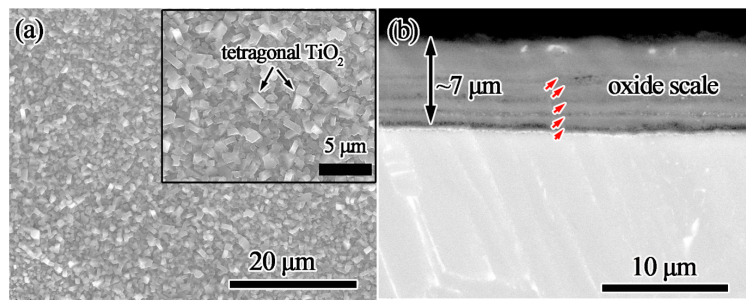
Surface (**a**) and cross-sectional (**b**) morphology of the bare Ti60 alloy after oxidation at 800 °C for 100 h.

**Figure 5 materials-13-05085-f005:**
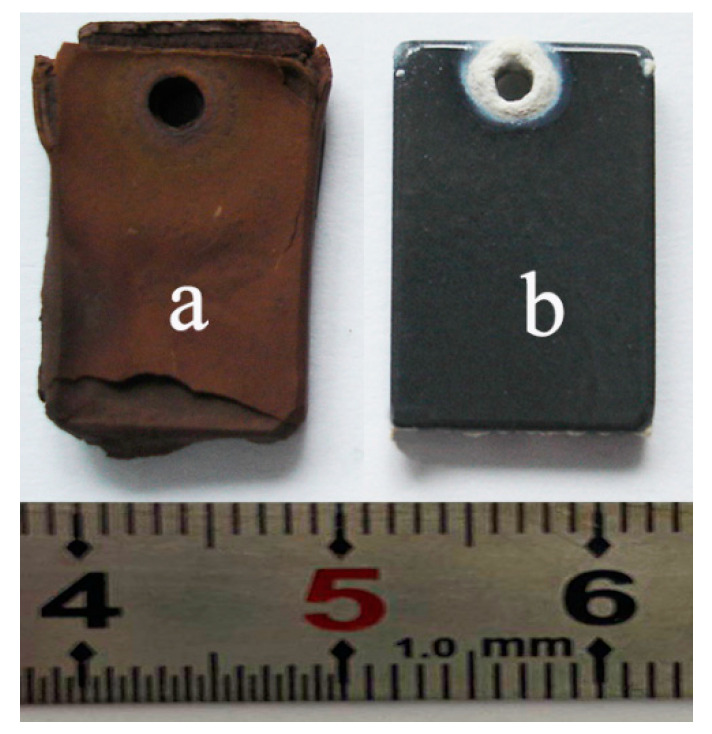
Macroscopic morphology of the substrate (**a**) and coated (**b**) sample after oxidation at 900 °C for 100 h.

**Figure 6 materials-13-05085-f006:**
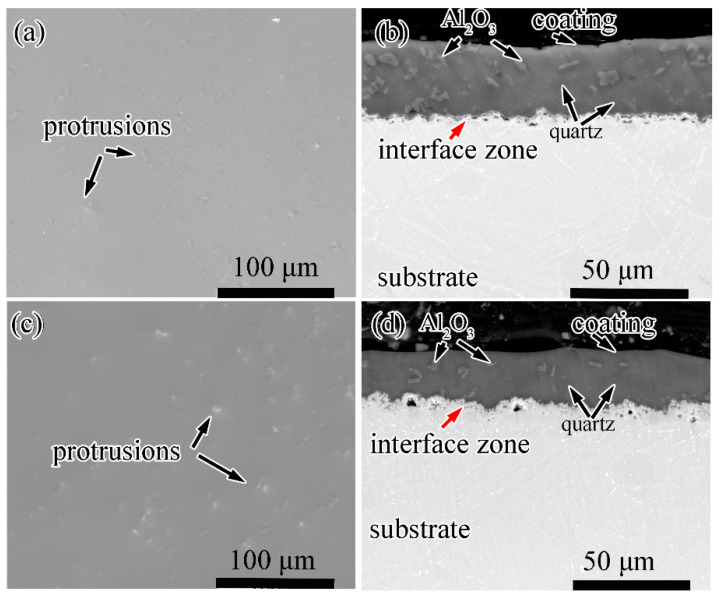
Surface and cross-sectional morphology of the Ti60 with composite coating after oxidation at (**a**,**b**) 800 °C and (**c**,**d**) 900 °C for 100 h.

**Figure 7 materials-13-05085-f007:**
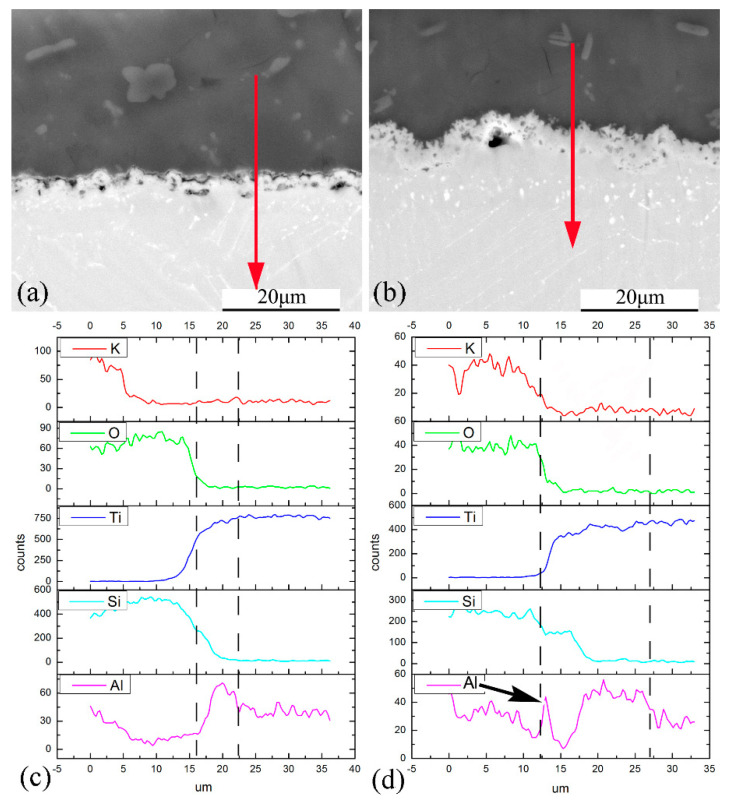
Elemental distribution of coating/alloy interface after oxidation at 800 °C (**a**,**c**) and 900 °C (**b**,**d**) for 100 h by EDS line-scan.

**Figure 8 materials-13-05085-f008:**
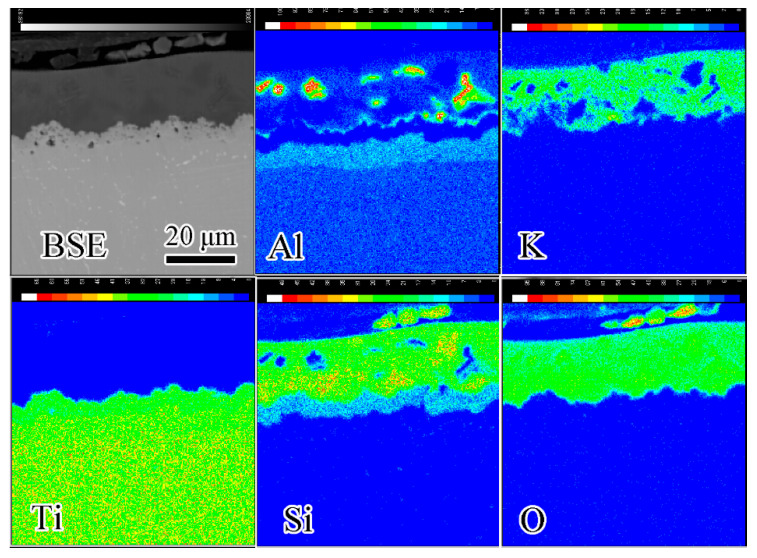
Elemental mapping of coating/alloy interface after oxidation at 900 °C for 100 h by electron probe microanalysis (EPMA).

**Figure 9 materials-13-05085-f009:**
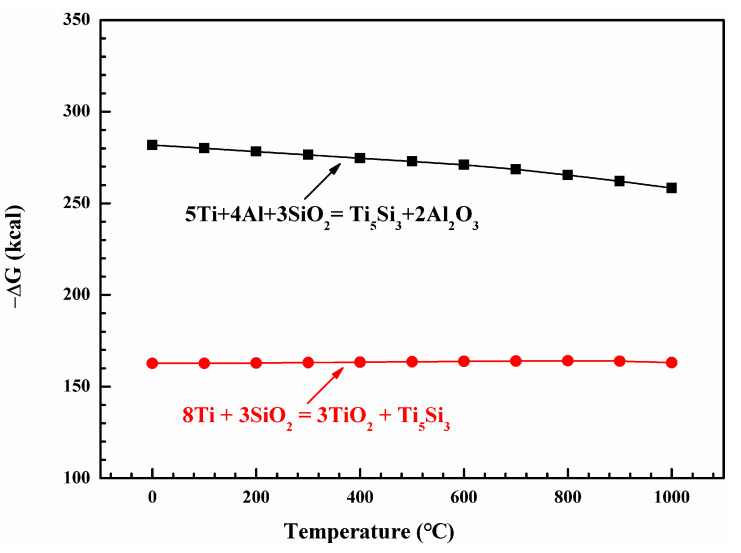
Standard Gibbs free energy changes (ΔG^θ^) of forming Ti_5_Si_3_ as a function of temperature (standardized to forming 1 mol Ti_5_Si_3_).

**Figure 10 materials-13-05085-f010:**
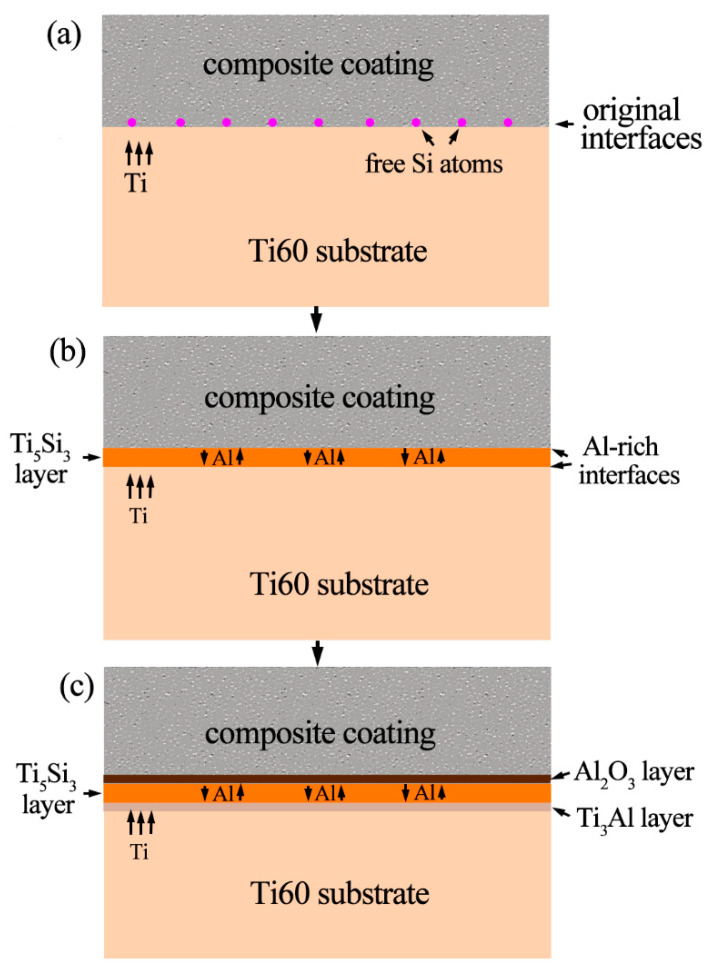
Schematic diagram showing the interfacial reaction between the coating and alloy at high temperature. (**a**) gathering of Si at the interface; (**b**) Al enrichment; (**c**) formation of the multi-layered structure.

**Table 1 materials-13-05085-t001:** Nominal composition of the substrate (wt.%).

Elements	Al	Sn	Zr	Mo	Nd	Si	Ti
Nominal Composition	5.6	4.8	2.0	1.0	1.0	0.3	Balance

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
