# Peer review of "Effect of SiO_2_–Al_2_O_3_ Glass Composite Coating on the Oxidation Behavior of Ti60 Alloy"

_materials, 2020, doi:10.3390/ma13225085_

Round 1
Reviewer 1 Report
I have following comments for the present manuscript:
1// Please mention the reasons for a choice of baking schedule (was it through previous experiences or due to some literature). Please discuss in detail.
2// Please include errors in Figure 2.
3// Please provide reference data in Figure 3 to compare the XRD patterns.
4// Please discuss the direction forward to the research for a potential reader.
Author Response
Reviewer 1:
The manuscript describes innovative research on oxidation and the role of protective coating. The state-of-the-art is sufficient, the methodology has only some gaps, results are properly shown, the discussion is interesting. However, I have several remarks about some details, which should be improved before publishing the paper. The most important: Results should only demonstrate the observations, Discussion is sometimes unclear and unjustified, Conclusions should be seriously improved.
(1) Line 13: the coating is composed of the aqueous solution?
Answer: The original material of the coating are aqueous solution of potassium silicate, alumina and quartz powders. After being solidified at high temperature, the aqueous solution of potassium silicate dehydrated and condensed to form potassium silicate glass. Thank you for your suggestion. It has been revised in this paper (Line 13-14).
.
(2) Lines 25-27, 98, 116, 236: a plural form (titanium alloys…) should be applied.
Answer: Thank you for the reminding. The plural form have been applied for the above lines. After a care inspection, the plural form have also applied for Line 31, 213, 245.
(3) Line 39: what types of crystals? Unclear.
Answer: The coarse needle-like crystals trend to precipitate in glass easily, which will cause stress concentration and is harmful for the mechanical properties of the coating. Therefore, the coarse needle-like crystals are undesirable. Relevant explanation has been added in the paper (see Line 41 in the revised manuscript)
(4) Line 46: space is necessary.
Answer: Thanks. Space has been added.
(5) Line 47: crystallization of glasses? The glasses mean a non-crystalline form of materials.
Answer: Yes, you are right that glasses is a non-crystalline. However, when it is baked or exposed at high temperature for long time, it becomes metastable and it is possible to precipitate crystallized phase. This is especially obvious when ceramics are added in the glass, which has been reported in Chen’s work (M. Chen, S. Zhu, F. Wang, Crystallization Behavior of SiO2–Al2O3–ZnO–CaO Glass System at 1123–1273 K, Journal of the American Ceramic Society, 93 (2010) 3230-3235).
(7) Line 53: small colloids? Small colloidal particles, I presume.
Answer: Sorry. It’s small colloidal particles and revision has been made.
(8) Lines 68, 73, 92, 114-115, 167: the Past Tense is recommended.
Answer: Suggestions have been accepted gladly.
(9) Line 80: a type of furnace and manufacturer should be given.
Answer: The type (a batch-type muffle furnace) of the furnace and manufacturer (Luoyang Kere Furnace Co., Ltd) have been added in Line 90-91.
(10) Line 93: what is the phase constitution?
Answer: The composite coating is composed of glass matrix, ceramic particles like Al2O3, SiO2 and products that formed during baking and oxidation. Therefore, there are different phases in the coating. Moreover, the types of the phase will also varies during baking and oxidation. In order to figure out the types of the phases of the coating after both baking and oxidation, XRD was used to analyzing the phase constitute of the coating.
(11) Lines 102, 113, 220: much lower, I suggest.
Answer: Suggestions have been accepted. (see Line114, 127, 240 in the revised manuscript)
(12) Lines 102-105, 119-121, 127-131, 143-144: The shown sentences should not appear in the Results section, but the Discussion.
Answer: Line 102-105 (Line 116-118 in the revised manuscript) was aimed to compare the results in the current work with Han’s work, which indicates that a lower oxidation rate was achieved for the current coating. It will be more convenient to make the comparison when these lines appear in the Results section.
Line 119-121(Line 132-134 in the revised manuscript) analyzed the possible reason for the formation of Ti5Si3 and Ti3Al in the XRD results.
Line 127-131 (Line 142-144 in the revised manuscript) compare the current results with Holmquist, which proves the validity of the current work.
Line 143-144 (Line 158-159 in the revised manuscript) displays the characteristics of the protective oxide scale, which explains why the oxide scale on bare Ti60 does not have the protection.
Based on the above explanation, all these sentence are helpful for the readers to understand the results in the current work well. Therefore, we would rather to keep the original position of them.
(13) Lines 106-108: Unclear, no logical relationship between the first and second sentences.
Answer: Sorry to have caused this confusing. A revision have been made as “It is well accepted that the oxidation rate of the coating increases with the elevating of the exposure temperature. However, the fact is that the composite coating in the current work showed a similar oxidation rate with phosphate-ceramic coating even exposure at higher temperature. Therefore, the composite coating here possesses superior oxidation resistance than phosphate-ceramic coating” (Line 118-123 in the revised manuscript).
(14)Line 109: These sentences should be joined.
Answer: Revision has been made as “When the oxidation temperature is elevated to 900 °C, the mass gain for both samples increases significantly” (Line 123 in the revised manuscript)
(15)Line 145: …gives protection … suggested.
Answer: Thank you. Suggestion has been accepted (Line 159 in the revised manuscript).
(16)Line 165: can characterize… suggested.
Answer: Thank you. Suggestion has been accepted (Line 186).
(17)Lines 199, 200-201, 215, 216-217, 219-220: please give references for these assumptions.
Answer: Thank you for your suggestions. Three references have been added, which is references 30, 31 and 33 in the revised manuscript: Ref. 30 (European Ceramic Society, 10 (1992) 347-367) for Line 199 (Line 221 in the revised manuscript), Ref. 31 (Geochimica et Cosmochimica Acta, 46 (1982) 2293-2299) for Line 200-201(Line 222-223 in the revised manuscript) and Ref. 32 (Introduction to the high temperature oxidation of metals, Cambridge University Press, 2006, pp 82-98) for Line 215, 216-217,219-220 (Line 236 and 238-241 in the revised manuscript).
(18)Line 214: were there any measurements for the first few seconds?
Answer: To the authors’ best knowledge, the mass gain in the first few seconds is hard to measure accurately. The oxide scale after oxidation for a few seconds is possible to be observed by TEM.
(19)Lines 223-224, 249: justify this assumption.
Answer: Sorry, we did not find the assumption in Line 223-224 (there is no assumption in Line 245-256 in the revised manuscript). For the assumption (occurrence of reaction 2) in Line 249 (Line 263 in the revised manuscript), it is hard to justify it directly. However, we have proved it to be reasonable by two aspects indirectly: first, both the products of Ti5Si3 and Al2O3 are observed at the interface by XRD, SEM and EPMA, which means reaction with the aforementioned products has occurred already; second, the standard Gibbs free energy changes of reaction 2 is negative, which proves that reaction 2 is thermodynamically feasible.
(20) Lines 224, 225: please write the full name of CTE when for the first time.
Answer: Revision has been made (Line 243 in the revised manuscript)
Lines 239, 266: improper phrase.
Answer: Phrase in this section has been check and revised (see Lines 259, 284-285 in the revised manuscript)
(21) Lines 243, 246, 254: why Reaction begins with a capital letter?
Answer: Sorry to have caused this confusing. Revisions have been made (Line 263,266, 270 and 274 in the revised manuscript)
(22) Line 274: justify that the coatings are dense. Besides, the conclusions should answer to the question, why. The explanation of good adherence has been already given, it is suggested to repeat it.
Answer: The surface morphology of the coating after baking has been added as Fig. 1b It can be noted from Fig. 1b that the coating are dense without any visible cracks or voids. Moreover, the coating are dense also from the cross-sectional morphology in Fig.6, Fig. 7 and Fig. 8. The first conclusion has been revised as “The composite coating exhibits a dense structure. It is adherent to the Ti60 substrate due to their similar CTE and occurring of interfacial reaction at the coating/substrate interface”
(23) Lines 276-277: the remark as above, please shortly describe the reason.
Answer: Revision has been made as above.
(24) Line 278: as above.
Answer: Revision has been made as above.

Reviewer 2 Report
The authors report the effect of composite coating on the oxidation behavior of Ti60 alloy. The introduction explains the work well and provides a clear background. However, there are missing details in the experimental section which I think would be important to describe. Additionally, the presentation of the data could be significantly improved. Here are the comments:
- ASPS abbreviation is not needed in the abstract, however, it should be introduced in the main manuscript at Line 68. There are also several abbreviations in the abstract which should be written in their full descriptions.
- Line 72 – how was the mixing process performed?
- Line 73-74 – how was the spraying process done? What type of spray and equipment is used? How long was it sprayed for and how is it controlled for the same thickness?
- Coating preparation step has quite a complex baking schedule. I think it may be beneficial to add a brief description and what is the purpose for each of the temperature steps.
- There were 3 samples tested to get average mass change. How is the consistency of the values? Maybe the authors could add a confidence interval for the data points in Figure 2 or describe the range of values between the samples.
- Is the substrate uniformly coated? I would suggest SEM images before the oxidation process to show the coating morphology prior.
- Line 134 – Authors stated cubic TiO2 as the typical morphology of TiO2. TiO2 is usually found in its tetragonal state, rather than cubic, and the morphology in Figure 4a does look more rectangular rather than cubic. Cubic TiO2 is really unusual to form and requires high temperatures and high pressures to achieve. It may be beneficial to either cite some reference work or add some XRD data collected from this sample to identify the phase. Please add the reference for the Ti-O graph which was discussed as well.
- Line 145 – The authors mentioned that the spallation has occurred and can be identified by naked-eyes. Would it be possible to add photographs of the samples?
- There are numerous statements without proper references in the discussion section, especially the comparison of diffusion coefficients between the various materials.
- Line 233 – Ti5Si3 “and” Ti3Al
- Why are the temperatures 800 and 900deg C chosen for the oxidation tests?
- The conclusion included “composite coating exhibits a dense structure and good adhesion to the substrate”. I would suggest to add “after oxidation tests”.
- What are the challenges using such a coating technique and what could be further optimized for improved oxidation resistance? Please add this into the discussion.
Suggestions to improve figure quality:
- Figure 2(b) inset is too low quality, the text could not be read.
- The highest intensity peak of the 800C 100h exceeds the plot scale of Figure 3. Please adjust the axis to ensure the entire plot fits in the box. Also, the “prepared” label should be “as-prepared”.
- I would suggest to label the substrate in Figure 4(b), similar to Figure 5(b) and (d). The scale bars in Figure 5 are not quite visible.
- Figure 7 elemental mappings are rotated sideways and inconsistent with the other figures. It would be better to keep consistent with the orientation in other figures. 18um is also a very awkward scale bar, I would recommend 20 um. The color scale is also not-visible.
Author Response
Reviewer 2:
Comments and Suggestions for Authors
The authors report the effect of composite coating on the oxidation behavior of Ti60 alloy. The introduction explains the work well and provides a clear background. However, there are missing details in the experimental section which I think would be important to describe. Additionally, the presentation of the data could be significantly improved. Here are the comments:
(1) ASPS abbreviation is not needed in the abstract, however, it should be introduced in the main manuscript at Line 68. There are also several abbreviations in the abstract which should be written in their full descriptions.
Answer: Sorry. All the abbreviation in the abstract have been replaced by full descriptions.
(2) Line 72 – how was the mixing process performed?
Answer: Sorry to have missed this part. Revision have been made as “In order to make the particles homogeneous, the mixture was milled via ball milling for 15 min. The speed of the ball milling is 1500 r/min” (Line 77-78 in the revised manuscript)
(3) Line 73-74 – how was the spraying process done? What type of spray and equipment is used? How long was it sprayed for and how is it controlled for the same thickness?
Coating preparation step has quite a complex baking schedule. I think it may be beneficial to add a brief description and what is the purpose for each of the temperature steps.
Answer: A more detailed spaying process has been added (Line 80-81 in the revised manuscript). The purpose of the baking process is to expel the water in the coating. However, it is undesirable to expel the water fast via evaporation. This is because the gaseous H2O (g) will result a substantial stress in the coating and cracks will form. The gradual and complex baking schedule insures the water departure from the coating slowly, which avoids the formation of cracks in the coating. The corresponding description has also been added in the revised manuscript (Line 82-83 in the revised manuscript).
(4) There were 3 samples tested to get average mass change. How is the consistency of the values? Maybe the authors could add a confidence interval for the data points in Figure 2 or describe the range of values between the samples.
Answer: Thank you for your suggestion. The mass change of the three samples are quite consistent. According to the reviewer’s suggestion, the error bars have been added in both the mass gain curves in Fig. 2.
(5) Is the substrate uniformly coated? I would suggest SEM images before the oxidation process to show the coating morphology prior.
Answer: Suggestion has been accepted. The surface morphology of the coating after baking has been added in Fig. 1b. According to Fig. 1b, the substrate was uniformly coated by the coating. The coating is impact without any visible cracks or voids.
(6) Line 134 – Authors stated cubic TiO2 as the typical morphology of TiO2. TiO2 is usually found in its tetragonal state, rather than cubic, and the morphology in Figure 4a does look more rectangular rather than cubic. Cubic TiO2 is really unusual to form and requires high temperatures and high pressures to achieve. It may be beneficial to either cite some reference work or add some XRD data collected from this sample to identify the phase. Please add the reference for the Ti-O graph which was discussed as well.
Answer: We are sorry that we have made a mistake in describing the morphology of the TiO2. It is tetragonal and we have made the revision corresponding. Moreover, the reference (H. Okamoto, O-Ti (Oxygen-Titanium), Journal of Phase Equilibria and Diffusion, 32 (2011) 473) of the Ti-O graph has been added as Ref. 29 in the revised manuscript.
(7) Line 145 – The authors mentioned that the spallation has occurred and can be identified by naked-eyes. Would it be possible to add photographs of the samples?
Answer: Thank you for the advice. The macroscopical morphology of the substrate and coated sample after oxidation at 900 °C for 100 h have been added as Fig. 5.
(8) There are numerous statements without proper references in the discussion section, especially the comparison of diffusion coefficients between the various materials.
Answer: Sorry. Six new references have been added in Part 4.1 to support the discussion, especially the oxidation process and diffusion behavior of the Ti and Al.
(9) Line 233 – Ti5Si3 “and” Ti3Al
Answer: Thank you. Revision has been made.
(10) Why are the temperatures 800 and 900 deg C chosen for the oxidation tests?
Answer: In fact, poor high temperature oxidation resistance of titanium alloys and titanium aluminides restricts their more application as high-temperature parts. The highest tested oxidation temperature for titanium alloys and titanium aluminides is about 950 ℃ in the open literature (Journal of Alloys and Compounds, 685 (2016) 784-798). However, the alloy deteriorated fast even with various alloying elements. As a results, the limit oxidation temperature for titanium alloys is 800-900 ℃. In order to show the superiority of the composite coating, oxidation temperature for the current test was chose around the limit oxidation temperature of titanium alloys.
(11) The conclusion included “composite coating exhibits a dense structure and good adhesion to the substrate”. I would suggest to add “after oxidation tests”.
Answer: Thank you. Suggestion has been accepted gladly.
(12) What are the challenges using such a coating technique and what could be further optimized for improved oxidation resistance? Please add this into the discussion.
Answer: We think there are two challenges that need to be consider before using the coating. The first is how to produce the same impact and intact coating on large-sized workpiece rather than in small samples. The second is the long term corrosion behavior of coating in the molten salts or environment with Cl- content. In fact, the above two aspects are also what we will focus on in the next research.
In order to further improve the oxidation resistance of the coating, the diffusion rate of oxygen in the coating should been lowered, which can be achieved by optimize the content and the structure of the coating. A new part (Part 4.3) has been added to discuss the developing trends of SiO2-Al2O3-glass composite coating, which includes ways to improve the oxidation resistance also.
Suggestions to improve figure quality:
(13) Figure 2(b) inset is too low quality, the text could not be read.
Answer: Sorry. All the figures in Figure 2 have been replaced with high resolution images.
(14) The highest intensity peak of the 800C 100 h exceeds the plot scale of Figure 3. Please adjust the axis to ensure the entire plot fits in the box. Also, the “prepared” label should be “as-prepared”.
Answer: Thank you for your kind suggestions. Figure 3 has been replotted according to the suggestions.
(15) I would suggest to label the substrate in Figure 4(b), similar to Figure 5(b) and (d). The scale bars in Figure 5 are not quite visible.
Answer: Thank you for your suggestions. The substrate in Fig. 4b has been labeled. The scale bars in Fig. 5 (Fig. 6 in the revised manuscript) have also been adjusted.
(16) Figure 7 elemental mappings are rotated sideways and inconsistent with the other figures. It would be better to keep consistent with the orientation in other figures. 18um is also a very awkward scale bar, I would recommend 20 um. The color scale is also not-visible.
Answer: Thanks. The directions of the elemental mappings has been rotated to be consistent with other figures and the scale bar has also adjusted.

Reviewer 3 Report
The manuscript describes innovative research on oxidation and the role of protective coating. The state-of-the-art is sufficient, the methodology has only some gaps, results are properly shown, the discussion is interesting. However, I have several remarks about some details, which should be improved before publishing the paper. The most important: Results should only demonstrate the observations, Discussion is sometimes unclear and unjustified, Conclusions should be seriously improved.
Line 13: the coating is composed of the aqueous solution?
Lines 25-27, 98, 116, 236: a plural form (titanium alloys…) should be applied.
Line 39: what types of crystals? Unclear.
Line 46: space is necessary.
Line 47: crystallization of glasses? The glasses mean a non-crystalline form of materials.
Line 53: small colloids? Small colloidal particles, I presume.
Lines 68, 73, 92, 114-115, 167: the Past Tense is recommended.
Line 80: a type of furnace and manufacturer should be given.
Line 93: what is the phase constitution?
Lines 102, 113, 220: much lower, I suggest.
Lines 102-105, 119-121, 127-131, 143-144: The shown sentences should not appear in the Results section, but the Discussion.
Lines 106-108: Unclear, no logical relationship between the first and second sentences.
Line 109: These sentences should be joined.
Line 145: …gives protection … suggested.
Line 165: can characterize… suggested.
Lines 199, 200-201, 215, 216-217, 219-220: please give references for these assumptions.
Line 214: were there any measurements for the first few seconds?
Lines 223-224, 249: justify this assumption.
Lines 224, 225: please write the full name of CTE when for the first time.
Lines 239, 266: improper phrase.
Lines 243, 246, 254: why Reaction begins with a capital letter?
Line 274: justify that the coatings are dense. Besides, the conclusions should answer to the question, why. The explanation of good adherence has been already given, it is suggested to repeat it.
Lines 276-277: the remark as above, please shortly describe the reason.
Line 278: as above.
Author Response
Reviewer 3:
I have following comments for the present manuscript:
(1)Please mention the reasons for a choice of baking schedule (was it through previous experiences or due to some literature). Please discuss in detail.
Answer: The baking schedule is determined through trial-and-error. There is a lot of water within the coating after spray. These water must be expelled slowly and avoid two form H2O(g), or crack will generates in the coating. The purpose of choosing such a baking schedule has been added in Line 82-83. As a detailed discussion will be contained in our next work “Effect of baking on the microstructure and oxidation behavior of SiO2-Al2O3-glass composite coating”, we would rather do not discuss this part in the current work.
(2) Please include errors in Figure 2.
Answer: Suggestion has been accepted gladly.
(3) Please provide reference data in Figure 3 to compare the XRD patterns.
Answer: Sorry, the data in the work of Holmquist (Ref.28) were summarized in table as below. Therefore, it is can’t been added in Figure 3 to compare with the current results directly. However, from the table below, cristobalite do not form until the oxidation temperature is as high as 890 ℃. Therefore, it is safe to say that cristobalite is easier to form at high temperature.
The above table is cited from the work of Holmquist ( Journal of the American Ceramic Society, 44 (1961) 82-86).
(4)Please discuss the direction forward to the research for a potential reader.
Answer: Thank you for the suggestion. A new part (Part 4.3) has been added to discuss the developing trends of the SiO2-Al2O3-glass composite coating.

Round 2
Reviewer 2 Report
Thank you for the great work on the revisions. The paper is significantly improved from before. I just have a few minor comments:
Line 77 - ASPS is still not described in full length
Figure 4a still says cubic TiO2
Could you add the direction of the EDS line scan for Figure 7a and b?
Author Response
Reviewer 2:
Thank you for the great work on the revisions. The paper is significantly improved from before. I just have a few minor comments:
- Line 77 - ASPS is still not described in full length
Answer: Sorry, the full name of ASPS, that is aqueous solution of potassium silicate, has been added.
- Figure 4a still says cubic TiO2
Answer: Sorry. Fig. 4a has been modified.
- Could you add the direction of the EDS line scan for Figure 7a and b?
Answer: Thank you. Arrows have added in Fig. 7a and b, which pointed out the direction of line scan. The corresponding description has also been added in Line 188.
Reviewer 3 Report
Despite several remarks, I am satisfied that all of them have been either taken into account and the manuscript has been adequately revised, or the proper answers explaining the used constructs have been given. Therefore, I have not other comments.
Author Response
Reviewer 3:
Despite several remarks, I am satisfied that all of them have been either taken into account and the manuscript has been adequately revised, or the proper answers explaining the used constructs have been given. Therefore, I have not other comments.
Answer:Thank you very much.